# A novel assay to measure low-density lipoproteins binding to proteoglycans

Esmond N. Geh[1]*, Debi K. Swertfeger[1], Hannah Sexmith[1], Anna Heink[1], Pheruza Tarapore[2], John T. Melchior[2,3,4], W. Sean Davidson[2], Amy Sanghavi Shah[1]

1 Division of Endocrinology, Cincinnati Children's Hospital Medical Center & the Department of Pediatrics, University of Cincinnati College of Medicine, Cincinnati, Ohio, United States of America, 2 Center for Lipid and Arteriosclerosis Science, Department of Pathology and Laboratory Medicine, University of Cincinnati, Cincinnati, Ohio, United States of America, 3 Biological Sciences Division, Pacific Northwest National Laboratory, Richland, Washington, United States of America, 4 Department of Neurology, Oregon Health and Science University, Portland, Oregon, United States of America

* Esmond.geh@cchmc.org

## Abstract

### Background

The binding of low-density lipoprotein (LDL) to proteoglycans (PGs) in the extracellular matrix (ECM) of the arterial intima is a key initial step in the development of atherosclerosis. Although many techniques have been developed to assess this binding, most of the methods are labor-intensive and technically challenging to standardize across research laboratories. Thus, sensitive, and reproducible assay to detect LDL binding to PGs is needed to screen clinical populations for atherosclerosis risk.

### Objectives

The aim of this study was to develop a quantitative, and reproducible assay to evaluate the affinity of LDL towards PGs and to replicate previously published results on LDL-PG binding.

### Methods

Immunofluorescence microscopy was performed to visualize the binding of LDL to PGs using mouse vascular smooth muscle (MOVAS) cells. An in-cell ELISA (ICE) was also developed and optimized to quantitatively measure LDL-PG binding using fixed MOVAS cells cultured in a 96-well format.

### Results

We used the ICE assay to show that, despite equal APOB concentrations, LDL isolated from adults with cardiovascular disease bound to PG to a greater extent than LDL isolated from adults without cardiovascular disease (p<0.05).

**Data Availability Statement:** All relevant data are within the paper and its Supporting information files.

**Funding:** This work was supported by the following grants: NIH K23HL118132 and R01HL157260 to

ASS, Procter Scholarship to ASS and NIH R01HL67093 and R01 HL157260 to WSD. The funding organizations had no role in the study design data collection, analysis, or preparation of the manuscript.

**Competing interests:** The authors have declared that no competing interests exist.

## Conclusion

We have developed an LDL-PG binding assay that is capable of detecting differences in PG binding affinities despite equal APOB concentrations. Future work will focus on candidate apolipoproteins that enhance or diminish this interaction.

## Introduction

Multiple lines of evidence from human arterial specimens, cell culture studies, and animal models support the response to retention hypothesis of atherosclerosis. This model suggests that the binding of LDL to ECM-PGs is a key step in the development of atherosclerosis [1–3]. Within the arterial intima, vascular smooth muscle (VSMC) cells synthesize and secrete PGs into the surrounding ECM. They predominantly produce chondroitin sulfate proteoglycans (CSPG) which account for 65% of total PGs extracted from the aorta smooth cells PGs [4, 5]. PGs consist of a core protein covalently linked to negatively charged glycosaminoglycan (GAG) chains that can interact with LDL through ionic interactions with positive charged lysine and arginine residues on APOB. This interaction is influenced by the size, environment, and enzymatic modifications of LDL [6–11] as well as degree of sulfation of the GAGs [12]. In general, LDL has a higher affinity for PG binding in adults with atherosclerotic heart disease [13, 14], obesity, insulin resistance and type 2 diabetes [15, 16]. Thus, the degree to which LDL binds PGs may serve as a marker of atherosclerotic risk.

Given the potential importance of LDL association with PGs, several studies (reviewed here [17]) have aimed to determine whether *in-vitro* assays can accurately measure the degree of LDL-PG interaction and detect atherosclerotic risk in humans. The development of these assays has undergone many changes as tools to study LDL-PG binding become available and attempts have been made to optimize the assay for use in human studies. The initial studies used turbidity to measure the formation of insoluble complexes resulting from LDL-PG binding [18, 19]. Later, methods like gel mobility shift assays, affinity chromatography and ELISAs became prominent [20–22]. Most of these studies relied on the addition of radioactively labelled LDL to purified PGs (versican, biglycan etc) or GAGs extracted from tissues and immobilized on a solid support matrix, followed by quantification of the amount of bound LDL [3, 23, 24]. However, the reproducibility of these assays was hampered because of the limited availability of extracted PGs from human or animal tissues, differences in binding affinities of LDL towards individual PGs, and the composition of arterial intima, which also contains non-PG containing fibrous proteins as well as smooth muscle cells, can affect the LDL- PG interaction.

To address these challenges, we sought to develop a cell-based assay that simulates the physiological mixture of PG's commonly found in the vessel wall within a cellular environment, thus eliminating the need for radioactive labeling, individual PG extraction, or purification from cells. This assay utilizes mouse vascular smooth muscle (MOVAS) cell line in combination with an in-cell ELISA technique to detect bound LDL. While the in-cell ELISA technique has been widely used in immunological and other biological studies [25, 26] to investigate the presence, localization, and quantification of cell surface antigens, its application in studying LDL-PG interaction, to the best of our knowledge, has not been previously published. In the current study, we show that this assay is quantitative and reproducible and can be used to detect differences in the PG binding affinities of LDLs isolated from a cohort of adults with and without cardiovascular disease.

## Materials and methods

### Reagents and antibodies

Dulbeccos Modified Eagles Medium and DNAse were purchased from Fisher Scientific, Pittsburgh, PA. Rabbit anti-chondroitin sulfate antibody was purchased from Bioss Antibodies Inc.,Woburn, MA. Normal goat IgG, rabbit IgG and TrueView were purchased from Vector Laboratories, Burlingame, CA. CF488 donkey anti-rabbit IgG, CF594 donkey anti-goat and EverBrite™ mounting medium with DAPI were purchased from Biotium, Fremont, CA. Biotin- and HRP-conjugated goat anti-human Apolipoprotein B antibodies were purchased from Academy Bio-Medical, Houston, TX. Chondroitin sulfate monoclonal antibody (CS-56), horseradish peroxidase (HRP)-conjugated goat anti-rabbit IgG were obtained from Life Technologies, Carlsbad, CA. Heparan Sulfate antibody (clone F58-10E4), was purchased from AMSBIO, Cambridge, MA and Alexa Fluor 488 together with Alexa Fluor 594 conjugated Donkey Anti-Mouse IgM from Jackson ImmunoResearch Laboratories, West Grove, PA. Papain, L-cysteine, chondroitin sulfate and 1,9-Dimethyl-Methylene Blue (DMMB) were all purchased from Sigma-Aldrich, Milwaukee, WI. ELISA-Bright HRP substrate, 24 and 96-well plates were obtained from VWR International, Batavia, IL. Casein and Poly-L-Lysine from EMD Millipore, Temecula, CA; G418 (Geneticin) from InvivoGen, San Diego, CA and Proteinase K from New England BioLabs, Ipswich, MA.

### Cell culture and maintenance

Mouse vascular (aortic) smooth muscle (MOVAS) cell line was purchased from ATCC (Manassas, VA) and cultured in HyClone Dulbeccos Modified Eagles Medium/High glucose (DMEM) supplemented with 10% FBS, penicillin (100 units/mL), streptomycin (100 μg/mL) and 200 μg/ml of G418. The media was changed twice a week and the cells were split at a1:6 ratio when they reached 90% confluence. For In-Cell ELISA experiments, the cells were seeded at 25,000 cells/well in 96-well polystyrene plates and for immunocytochemistry, on glass cover slips in a 24-well format plate at 75,000 cells/well and grown in DMEM without G418. At 3–5 days after reaching confluency, the cells were harvested for experiments. Human ascending aorta vascular smooth muscle (HAAVAS) cells were purchased from Creative Bioarray, Shirley, NY.

### Study participants

The LDL-PG binding assay was tested in participants with and without cardiovascular disease. Participants were recruited as part of the Fairbanks Institute biorepository at Indiana University (NCT01386801, NCT00741416) between 2007–2010. Study participants with cardiovascular disease were recruited based on an electronic health record confirmed history of at least one of the following: angioplasty, with or without stent placement; coronary artery bypass graft surgery; diagnostic angiogram; or positive catheterization results showing $\geq$ 50% occlusion. Healthy adults (controls) were simultaneously recruited and had no confirmed history of cardiovascular disease (as defined above) or other risk factors (i.e., diabetes). Plasma was obtained from the parent study and stored at −80˚C until use. Samples were never thawed. The study was conducted in accordance with Indiana University's Internal Review Board (Protocol 1011003179: Multicenter Research Study to Build a Repository). Participants provided written informed consent for use of their samples and data.

### Lipoprotein isolation

LDL was isolated by sequential ultracentrifugation from donor plasma obtained from Hoxworth Blood Bank as previously described [27]. The density was adjusted to 1.019 g/mL with

potassium bromide, followed by centrifugation at 15 degrees Celsius in a 70Ti rotor at a speed of 360,000 g for 18 hours to float the very low density lipoprotein (VLDL) and chylomicrons. The bottom fraction was collected, adjusted to 1.063 g/mL, and centrifuged to collect LDL (1.019–1.063 g/mL), which was then dialyzed into phosphate buffer saline (PBS) buffer [10 mM PBS, 140 mM NaCl, (pH 7.4)]. Protein concentration was determined by bicinchoninic acid (BCA) method using Pierce BCA protein determination kit.

## Immunocytochemistry

MOVAS cells were grown on coverslips for 24 h, washed with PBS before incubating with 4% paraformaldehyde (PFA) for 20 minutes at RT. The coverslips were washed three times with PBS and then blocked with 3% casein in PBS-Tween 20 (PBS-T) for 1 h prior to incubation with 0.5 mg/mL LDL protein for 30 min. LDL was then removed, coverslips washed and incubated with goat anti-human apolipoprotein B antibody (1:500 dilution in 1% BSA in PBS-T) and rabbit anti-chondroitin sulfate antibody (1:250) was added simultaneously for 1 h. For isotype controls, goat and rabbit IgGs were used at similar concentrations as anti-APOB and anti-CS respectively. Coverslips were washed with PBS-T before adding CF594 conjugated donkey anti-goat and CF488 donkey anti-rabbit (both at 1:500 dilution) for an additional hour RT. Both secondary antibodies were diluted in PBS buffer containing 10% normal donkey serum and 1% BSA. The coverslips were then washed and incubated with TrueView (Vector Laboratories, Burlingame, CA) autofluorescence quencher for 10 min (we found the autofluorescence quenching step was essential for reducing background fluorescence). After quenching, the coverslips were again washed twice with PBS and mounted onto glass slides using Ever-Brite™ mounting medium with DAPI (Biotium, Fremont, CA). The slides were air-dried and photographed with an Olympus BX51 fluorescence microscope equipped with a 20X objective and a charge-coupled device camera. Confocal microscope images were acquired on an LCM710 microscope using plan-Achromat 63x oil immersion objective.

## Image analysis

Correlation analysis and image quantification were performed using ImageJ Fiji (http://fiji.sc). This version of ImageJ contains a pre-installed plugin, *Coloc2*, that can be used to obtain colocalization parameters such as Mander's correlation coefficient which is based on pixel intensities and correlation between two fluorescent images [28]. ImageJ Fiji was also used to quantify LDL binding.

## Extraction of Sulfated Glycoaminoglycans (GAGs)

The method for extracting GAGs from MOVAS cells was adapted with modifications from [29, 30]. Cells were grown in T75 culture flasks until 7 days post confluence. The cells were washed twice with PBS and the GAGs were extracted using papain extraction buffer (3 ml) containing 5 mg of papain in phosphate buffer (0.2 M sodium phosphate buffer, pH 6.4, 400 mg sodium acetate, 200 mg EDTA and 40 mg cysteine hydrochloride) at 60˚C for 3 h, after which the contents were transferred to 1.5 mL tubes. The tubes were centrifuged at 10,000g for 10 min to remove cellular debris and the supernatant transferred to new tube. Proteinase K (150 µg/mL) and DNAse (10 U/mL) were added and vortexed prior to overnight incubation at 37ºC. The next day, 200 µL of the sample was aliquoted into new centrifuge tubes; 1 mL of 1,9-Dimethyl-Methylene Blue (DMMB) complexing solution (8 mg DMMB, 1.52 g glycine, 0.8 g NaCl and 47.5 mL of 0.1 M acetic acid and brought to 500 mL volume with sterile distilled water and then filtered using a 0.45 µm filter) was added to each tube and vortexed for 30 min. The tubes were spun at 10,000g for 10 min and the supernatant decanted. 200 µL of DMMB

de-complexing solution (4 M guanidine hydrochloride, 10% propan-1-ol and 50 mM sodium acetate, pH 6.8) was added to the pellet and vortexed for 30 min. The final solution (suspended pellet and supernatant) was then pooled and 150 μL was used to determine GAG concentration by comparing the absorbance at 525 nm, read with a SpectraMax plate reader, to that of a standard curve generated by 0 to 50 μg/mL of chondroitin sulfate.

After determining the concentration of GAGs in MOVAS cells to be approximately 1 μg per well, the extracted GAGs and CS were also used at 1 μg per well to coat ELISA plates. The efficiency of coating was about 2% for CS and 25% for GAGs. Plate coating was achieved by incubating plates with 100 μL of 0.1% Poly-L-Lysine for an hour at room temperature (RT). Poly-L-Lysine was decanted and 100 μL per well of 10 μg/mL of extracted GAGs or CS, were then incubated at 37°C overnight.

### In-Cell ELISA (ICE) to measure LDL-PG binding

An in-cell ELISA to quantify APOB bound to proteoglycans was performed as described [31]. Post confluent MOVAS cells were fixed with 4% PFA for 20 min After fixation, the PFA was removed, and wells were washed with PBS. The plates were then blocked overnight with 200 μL/well of 3% casein in PBS-T. After overnight incubation, the wells were then emptied, and washed 4 times for 5 min with PBS-T. One hundred microliters of ultracetrifugally isolated LDL, at concentrations specified in individual experiments, were added to the wells, and incubated for 30 min RT. The wells were washed again, 4 times with PBS-T. One hundred microliters of 1:5000 dilution of HRP-conjugated goat anti-human apolipoprotein B antibody (in 1% BSA-PBST) were then added to the wells and incubated for 1 hour RT. The wells were washed, developed with chemiluminescent (ELISA-Bright) substrate, and immediately read using a Gen5 plate reader.

For batch preparation of cells, 20 plates containing the same number MOVAS cells were processed at the same time, after the blocking step, the plates were then air dried at RT, sealed, and kept frozen at -80°C until use. Each batch can be used for 6 months without the loss of LDL binding ability after which time, binding gradually declines.

### SDS-PAGE analyses

SDS-PAGE analyses was performed as described [32]. Five micrograms of protein were loaded per lane and electrophoresed under reducing conditions and the gels were subsequently stained with Coomassie Brilliant Blue (CBB).

### Size exclusion chromatography

To test an alternative form of LDL from that isolated by ultracentrifugation, we used size exclusion chromatography [33]. Three hundred and fifty-four microliters of plasma were size fractionated using a Single Superdex 200 increase column coupled with two Superose 6 increase gel filtration columns (10/300 GL; GE Healthcare), with a flow rate of 0.3 ml/min in PBS buffer on an ÄKTA FPLC system (GE Healthcare). Eluate was collected as 1.5-mL fractions on a Frac 900 fraction collector (GE healthcare) maintained at 4°C. Fractions corresponding to APOB containing LDL were collected, pooled, and concentrated using Amicon Ultra 4 mL centrifugal columns (MilliporeSigma, Burlington, MA).

### Statistical analysis

All data are reported as the sample mean ± standard deviation. A paired student's t-test was used to compare dose response differences when LDL was used from a single source,

otherwise, an unpaired student's t-test was used. A p-value of $< 0.05$ was considered statistically significant.

## Results

Mouse vascular smooth muscle cells (MOVAS) display a phenotype that is typical of smooth muscle cells in culture and have been frequently used as *in vitro* models for studying vascular calcification and LDL cholesterol metabolism [34–36]. They express a wide variety of proteoglycans (PGs) including those containing heparin, dermatan and chondroitin sulfate [37]. Proteoglycans and their associated glycosaminoglycans (GAGs) form the ground substance where fibrous proteins attach to form the extracellular matrix (ECM). Chondrotin sulfate (CS) and heparin sulfate (HS) PGs, which account for the majority of PGs in the aorta wall, are the major proteoglycans in MOVAS cells. CS has a high affinity for LDL, and was found to be associated with APOB in the extracellular space of injured aortas [38]. Moreover, LDL has also been reported to co-localize with PG in the ECM [39, 40]. Thus, we asked if LDL co-localizes with CS and HS in MOVAS cells.

Fixed sub-confluent (approximately 20%) MOVAS cells were incubated with UC-LDL, followed by staining with antibodies against APOB, CS, and HS. Confocal microscopy was used to capture a series of z-stack images, as shown in Fig 1. The APOB fluorescence (red) observed in these cells was primarily localized extracellularly, as shown in Fig 1A and 1D. The fluorescence (green) patterns of CS and HS displayed a similar distribution pattern (Fig 1B and 1E). When both images were merged, the regions exhibiting equal fluorescence from both channels appeared yellow, as highlighted in the inserts of Fig 1C and 1F. Similar results were obtained when human ascending aorta cells (HAAVAS) were used (S1 Fig in S1 File).

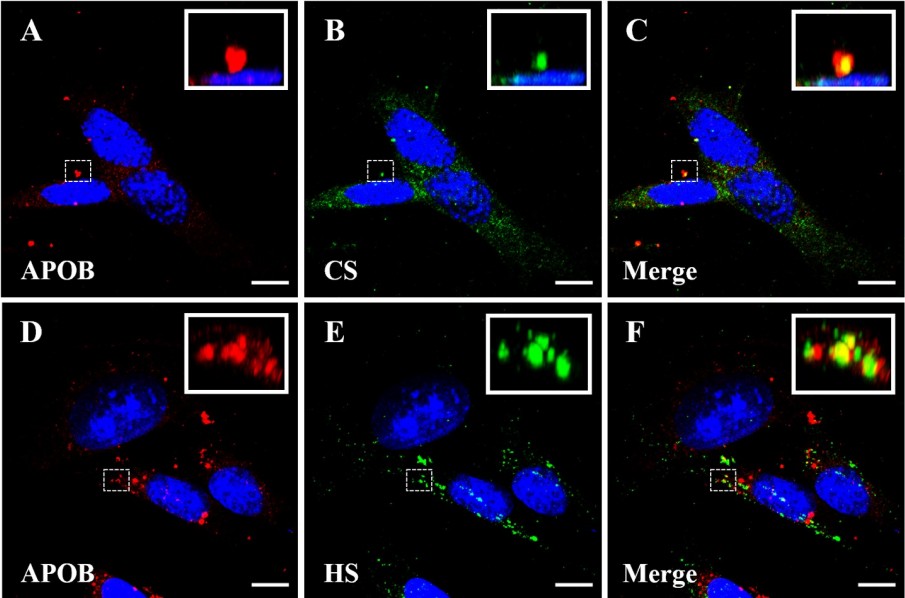

**Fig 1. LDL colocalizes with CS and HS proteoglycans in MOVAS cells.** Maximum intensity projection (MIP) of a z-stack images of MOVAS cells labeled with APOB (**A, D**), *red*; CS (**B**), and HS (**E**), *green*. DAPI, used to stain nuclei, is shown in *blue*. **C** and **F** represent merged MIP images for CS and HS, while *inserts* in the upper right display an enlarged 3D rendition of regions of interest (*dotted box*) within the respective images. Colocalized regions are shown in yellow. Scale bars indicate 10 μM. *Abbreviations; CS-chondroitin sulfate, HS-heparan sulfate.*

Our colocalization experiments using sparsely seeded cells show that LDL binds mainly to the ECM of MOVAS and co-localizes with CSPG and HSPGs. Based on this observation, we asked whether LDL would bind to confluent MOVAS cells in a similar manner, since this would more likely represent the *in vivo* state. Indeed, using confluent monolayer cells, APOB fluorescence was more pronounced in the pericellular and extracellular matrix areas forming a bright meshwork structure in between the cells with areas of dimmer fluorescence on the cell surface, Fig 2B. This pattern of fluorescence was enhanced in round mitotic cells (Fig 2B, *white arrows*) where the APOB signal was strongest, giving a ring- or donut-shaped appearance suggesting that dividing cells produce pericellular material that has high affinity for LDL binding. Chondroitin sulfate fluorescence was uniformly distributed on the surface of the cell (Fig 2C). These data illustrate that LDL is binding predominantly to CS in the ECM, with less LDL binding to cell surface CS. Colocalization analysis (see Materials and Methods) revealed a Mander's overlap coefficients of 48.8% for APOB signal overlapping with CS and 39.2% for CS signal overlapping with APOB.

To estimate the fraction of LDL that is associated with the pericellular and extracellular matrix areas versus the cell surface, we performed a quantitative analysis of randomly selected cells. This was done by calculating the intensity of fluorescent signal of a given cell surface (S) and the value was subtracted from the total intensities (T) of the cell surface plus extracellular fluorescence. The final value obtained (T-S) was divided by T then multiplied by 100 to obtain a percentage. This analysis revealed that 70–95% of the total LDL in the cells are located either pericellularly or within the ECM. The highest values were obtained from the smaller, round mitotic cells, which consistently gave a value above 90%. This analysis was repeated by a second experimenter giving similar results. The results suggest that almost half of the LDL particles (48.8%) bound to the cells are associated with CS and most of the LDL (70–95%) is

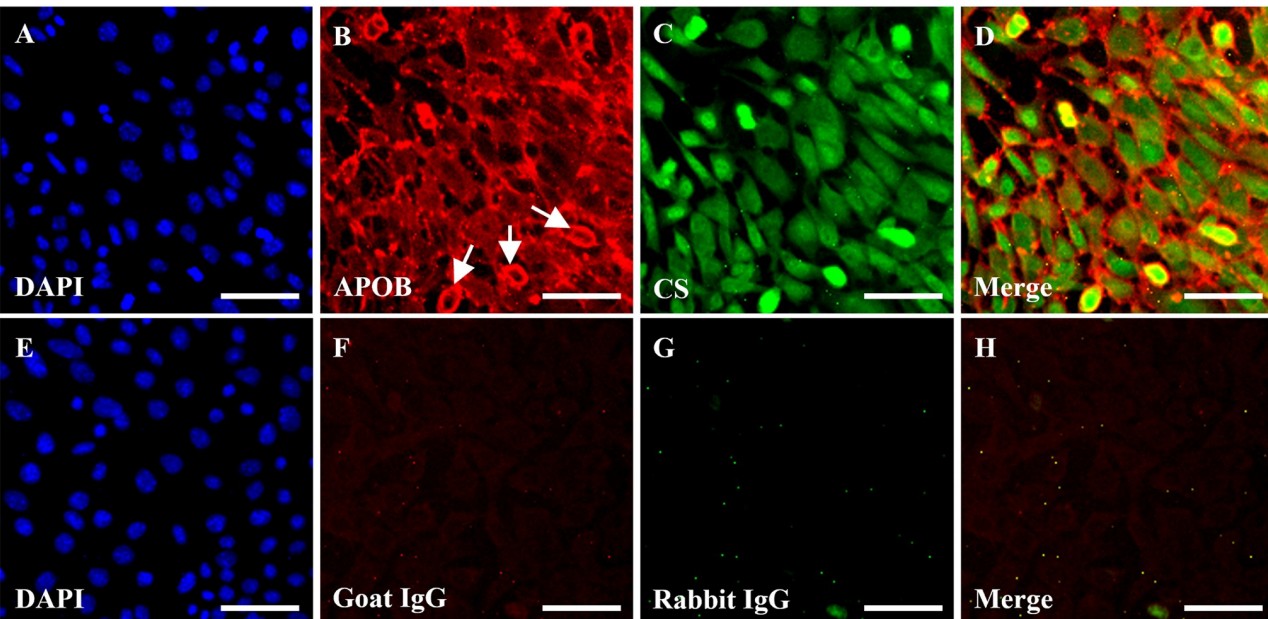

**Fig 2. LDL binds to the ECM of mouse vascular smooth muscle (MOVAS) cells.** Immunofluorescence staining for LDL and chondroitin sulfate in monolayer MOVAS cells. The cells were fixed with 4% PFA, blocked with 3% casein and incubated with 0.5 mg/mL of LDL for 30 minutes. Immunostaining was performed using goat anti-human APOB **(B)** and rabbit anti-CS **(C)** primary antibodies followed by CF594 donkey anti-goat (*red*) and CF488 donkey anti-rabbit (*green*) secondary antibodies. Negative control cells **(E-H)** were incubated with normal goat **(F)** and rabbit **(G)** IgGs using similar concentrations as anti-APOB and anti-CS respectively. Nuclei (*blue*) were stained with DAPI **(A, E)**. Scale bars, 50 μM.

associated with the pericellular or extracellular matrices. These data indicate that LDL binds to both cell surface PGs (CS) as well as other constituents (possibly other PGs) in the ECM. Given these findings, we hypothesized that MOVAS cells could be used as an *in vitro* model for measuring LDL binding to the ECM-PGs.

To test this hypothesis, we measured LDL binding to ECM-PGs in MOVAS cells cultured in a 96 well plate format. This was achieved by performing an in-cell ELISA using a goat HRP-conjugated antibody raised against human APOB. Two negative controls were included; one is antibody specificity control (wells with cells, without LDL) and the other is to control for LDL binding to the plastic plates (wells without cells, with LDL). Less than 2% of LDL binds to the plate without cells. The binding of LDL to plates was considered background and was subtracted from experimental values to obtain LDL-PG binding. In instances where we tested several concentrations of LDL, each concentration had a corresponding empty well.

Fig 3A shows that LDL binds to MOVAS cells in a dose dependent manner. A similar binding pattern was seen in wells coated with GAGs extracted from the cells and purified CS (Fig 3B). To determine the specificity of LDL binding to ECM-PGs, we performed a competitive binding assay that measures the binding of biotin-labeled LDL pre-incubated with increasing concentrations of unlabeled LDL prior to adding to the cells. As seen in Fig 3C, biotinylated LDL binding to MOVAS cells was reduced with increasing concentrations of unlabeled LDL.

To determine if LDL binding to the cells is dependent on CS, we pre-treated MOVAS cells or their extracted GAGs and CS with α-CS antibodies. LDL binding to cells was reduced with increasing α-CS antibody concentrations and at the highest α-CS concentration, there was a 28.5% decrease in LDL binding (Fig 4A). This effect was more pronounced when similar amounts of GAGs or CS were used to coat the plates (Fig 4B). The fact that significant LDL binding occurred in the presence of excess α-CS antibody suggests potential additional binding sites on the cells, consistent with the immunocytochemical analysis in Fig 2. To ensure that the observed binding was not due to the LDL receptor (Ldlr) in MOVAS cells, we evaluated LDL binding in cells that had been pre-treated with either α-CS or α-Ldlr antibodies. While pre-treatment with α-CS antibodies reduced LDL binding, pre-treatment of cells with α-Ldlr antibodies did not, suggesting Ldlr was not a major factor in our observed LDL association with the cells (Fig 4C). We cultured MOVAS cells in the presence or absence of increasing concentrations of sodium chlorate, an inhibitor of endogenous proteoglycan sulfation,

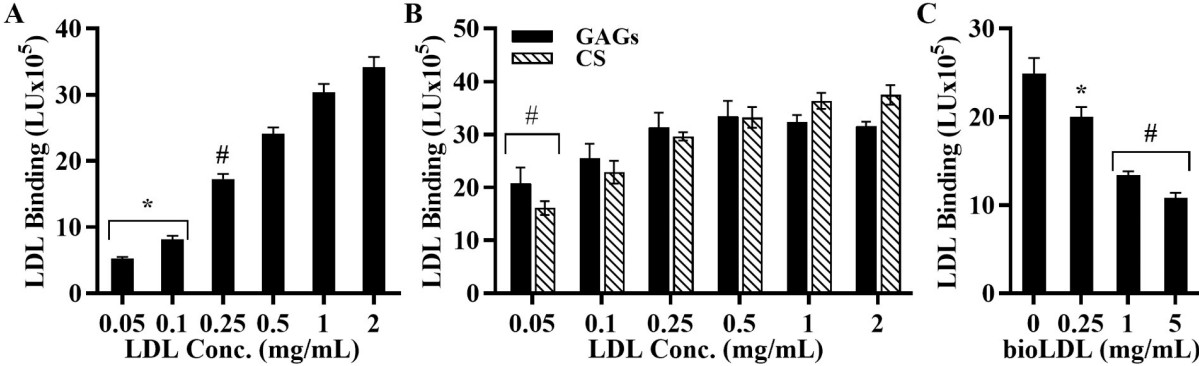

**Fig 3. LDL binds ECM in a dose dependent and specific manner. (A)** MOVAS cells and **(B)** GAGs extracted from MOVAS (*black bars*) and chondroitin sulfate, CS (*grey bars*) were preincubated with various concentrations of LDL. **(C)** Biotin-labeled LDL (0.5 mg/mL) was pre-incubated with increasing concentrations of unlabeled LDL (cold LDL). Results are expressed as LDL binding in light units. Bound LDL signal was calculated by subtracting the background signal (empty wells with corresponding amount of LDL) from the experimental signals (wells with cells). The mean and standard deviation (n = 4) are shown. Paired T-test show a significant difference (*p<0.01, #p<0.001) between LDL treated and untreated samples.

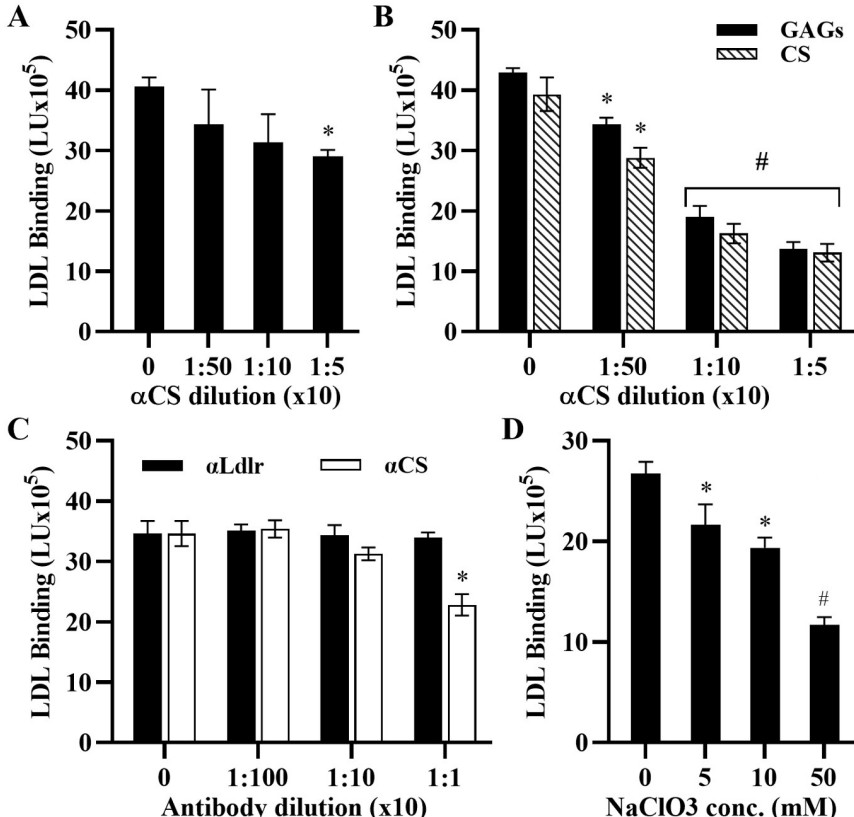

**Fig 4. LDL binding is mediated by sulfated PGs. (A)** MOVAS cells and **(B)** GAGs extracted from MOVAS (*black bars*) and chondroitin sulfate, CS (*grey bars*) were preincubated overnight with dilutions of chondroitin sulfate (α-CS) antibody prior to adding 1 mg/mL LDL. **(C)** Cells were pre-incubated overnight with various dilutions of antibodies against mouse LDL receptor (α-Ldlr) and chondroitin sulfate (α-CS) before adding 1 mg/mL LDL. **(D)** Cells were serum starved for 12 hours prior to incubation with varying concentrations of sodium chlorate (NaClO3) for 24 hours followed by incubation with 1 mg/mL LDL. Results are expressed as LDL binding in light units (LU). Bound LDL signal was calculated by subtracting the background signal (empty wells, LDL bound to plates) from the experimental signals (wells with cells). The mean and standard deviation (n = 4) are shown. Paired T-test show a significant difference (*p<0.01, #p<0.001) between LDL treated and untreated samples.

previously shown to suppress sulfation in MOVAS cells [41]. Fig 4D shows that, after 24 h in culture, dose dependent loss of sulfation caused a decrease in LDL binding to MOVAS cells. Taken together, these results suggest that a majority of the binding of LDL to MOVAS cells is mediated by negatively charged sulfate groups on PGs and depends on LDL concentration.

Previous work by others have shown that in an *in vitro* system, the binding of LDL to PGs is dependent on the pH of the milieu [42, 43]. Therefore, we evaluated the ability of pH to affect LDL binding in our cell culture system. We incubated MOVAS cells with 1 mg/mL of LDL protein diluted in PBS from pH 4 to pH 9. As shown in Fig 5A LDL-PG binding was reduced as the pH increased, indicating that changes in pH, as seen *in vivo* [44, 45], affects binding of LDL to PG. Similar results were obtained when LDL diluted in buffer pH 4–9 was tested on GAGs extracted from MOVAS cells and CS alone (Fig 5B).

Next, we tested the reproducibility of the assay using ultracentrifugally isolated LDL from the plasma of a single donor. Five experiments were performed on different days within a month using the same antibody, same batch of cells with each sample run in quadruplicates. Table 1, shows the intra-assay coefficient of variation ranged from 2.9–7.2%, while the inter-

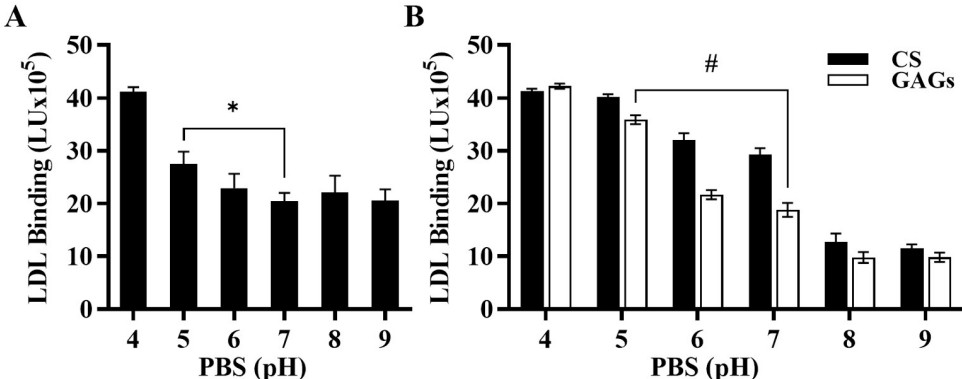

**Fig 5. Low pH enhances LDL binding to ECM. (A)** MOVAS cells and **(B)** GAGs extracted from MOVAS (*black bars*) and chondroitin sulfate, CS (*grey bars*) were incubated with 1 mg/mL LDL in PBS at pH 4–9. Results are expressed as LDL binding in light units (LU). Bound LDL signal was calculated by subtracting the background signal (empty wells, LDL bound to plates) from the experimental signals (wells with cells). The mean and standard deviation (n = 4) are shown. Paired T-test show a significant difference (*p<0.01, #p<0.001) between LDL treated and untreated samples.

assay coefficient of variation (CV) within the 5 experiments was <10%. We thus concluded that the assay is sensitive and reproducible.

The LDL-PG binding was further evaluated using LDL isolated by ultracentrifugation from 5 donor individuals. The APOB content of each individual's LDL was determined by a clinical analyzer and an equal amount (200 μg/mL) of APOB was used to assess LDL-PG binding abilities. As seen in Fig 6A, the PG binding (Fig 6B) differed among individuals suggesting the characteristics of the LDL particles may be important for its affinity towards PGs. We tested 5 LDLs from another set of individuals whose LDL was isolated by an alternative method, gel filtration chromatography which separates LDL by size (labeled LDL I-IV) as opposed to density, and results again show that using equal amounts of APOB (Fig 6C) there are differences in LDL ability to bind PG among individuals (Fig 6D). The use of gel filtration chromatography gave us an opportunity to evaluate LDL particle size on PG binding. LDL subfraction I represent LDL particles that are larger in size compared to LDL subfraction IV (Fig 6E). Fig 6F shows the subfraction containing smaller LDL particles exhibited greater binding to proteoglycans than the subfraction of LDL containing larger sized particles. Due to differences in size and gel migration patterns of the various LDL fractions, a western blot showing that the samples contain equal APOB, was performed on these fractions (S3 Fig in S1 File). These results indicate that not only can the assay detect differences in LDL binding to PG between individuals, but it can also detect differences between different sized LDL particles.

**Table 1. Intra- and inter-assay variability.**

| A-Intra-Assay Variability (n = 4) | | | | | |
|---|---|---|---|---|---|
| *Experiments* | *Day 1* | *Day2* | *Day3* | *Day4* | *Day5* |
| Average OD (x10⁵) | 34.3 | 30.3 | 31.4 | 35.2 | 38.4 |
| StDev (x10⁵) | 2.3 | 2.2 | 0.9 | 2.2 | 2.6 |
| CV (%) | 6.8 | 7.2 | 2.9 | 6.2 | 6.7 |
| B-Inter-Assay Variability (n = 5) | | | | | |
| Average OD (x10⁵) | | Range (x10⁵) | | StDev (x10⁵) | CV (%) |
| 33.9 | | 30.3–38.4 | | 3.2 | 9.4 |

Abbreviations: OD-Optical Density; StDev-Standard deviation; CV-Coefficient of variation.

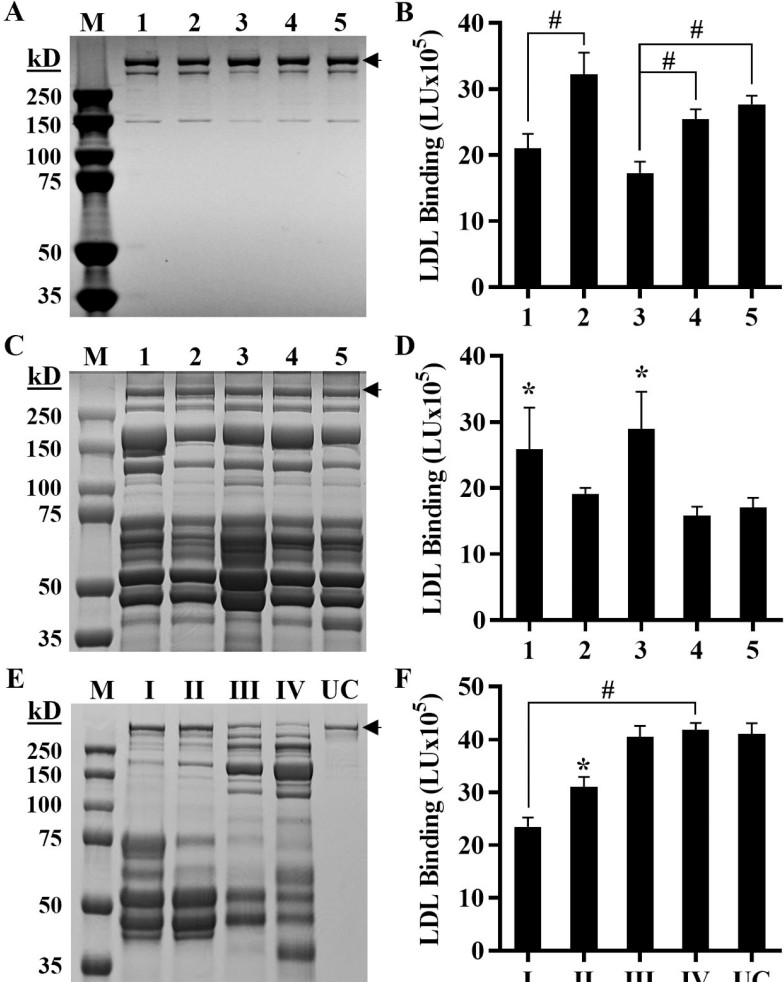

**Fig 6. The ICE assay detects differences in PG binding of various LDLs.** CBB stained gels of ultracentrifugally isolated LDLs **(A)**, FPLC purified LDLs **(C)** and LDL subfractions **(E)** [*left panels*] were used to perform an In-cell ELISA (B), (D) and (F), respectively [*right panels*]. The numbers 1–5 represent individual plasma donors, while I-IV represent FPLC isolated LDL fractions. Results are expressed as LDL binding in light units (LU). Bound LDL signal was calculated by subtracting the background signal (empty wells, LDL bound to plates) from the experimental signals (wells with cells). Black arrows indicate the expected location of APOB band. The mean and standard deviation (n = 4) are shown. Unpaired T-test show a significant difference (*p<0.05, #p<0.01) between indicated LDL signal versus the lowest LDL signal. *Abbreviations*: *CBB-Coomassie brilliant blue; FPLC-fast protein liquid chromatography; UC-ultracentrifugally isolated; Kd-kilodalton; M-Marker.*

Next, we asked whether the LDL-PG binding assay is sensitive enough to detect differences LDL-PG binding between a cohort of adults with and without cardiovascular disease. Twelve adults with cardiovascular disease (mean age 58.7 ± 3.7 years, mean BMI 30.4 ± 2.4 kg/m², and 67% female) and 10 controls without cardiovascular disease (mean age 54.4 ± 8.0 years, mean BMI 27.44 ± 2.74 kg/m², and 30% female) were studied. Due to limited plasma, the original assay (using HRP-labeled anti-human APOB) was slightly modified to use HRP-conjugated Streptavidin in combination with biotinylated anti-human APOB. The addition of a secondary antibody step increased the sensitivity by about 10-fold giving the assay the ability to detect differences in LDL-PG binding of about 2 μg/mL LDL (S4 Fig in S1 File).

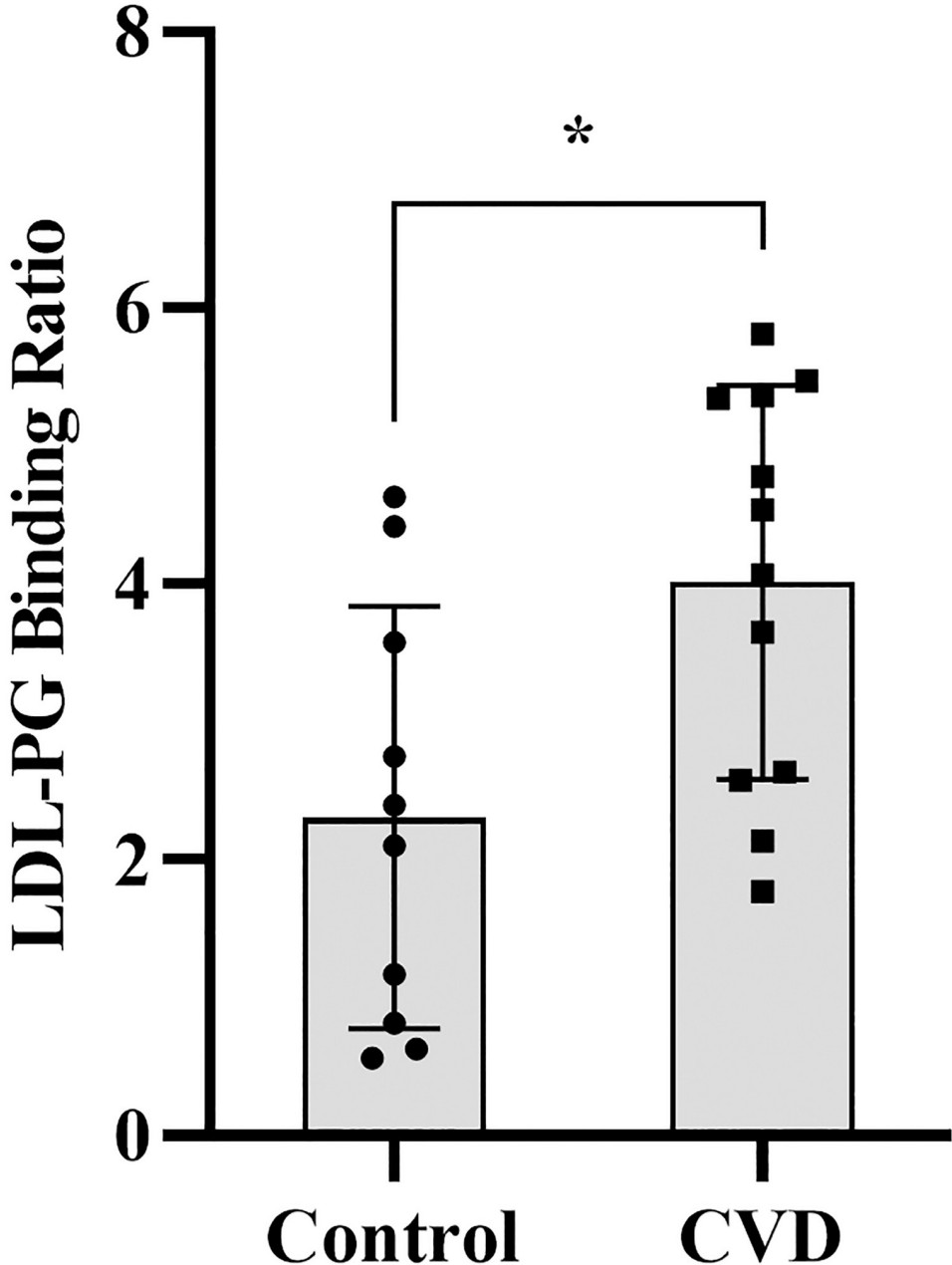

**Fig 7. ICE assay detects differences in LDLs from CVD patients and controls.** FPLC isolated LDLs from control and cardiovascular disease patients were used in an In-cell ELISA. The results are expressed as LDL binding ratio calculated by dividing the signal of each LDL with the signal generated by 10 μg/mL ucLDL within the same plate. The mean and standard deviation (n = 4) are shown. Unpaired Students *t*-test show a statistical significance difference of *$p < 0.05$ between controls and cardiovascular disease patients (CVD).

LDL was isolated from participants' plasma by gel filtration chromatography. APOB concentrations of each LDL was subsequently determined by clinical analyzer and adjusted to obtain 40 μg/Ml. A stock of ultracentrifugally isolated LDL was used at 10 μg/mL APOB as an internal control in every assay plate. LDL-PG binding assay was performed using equal APOB

concentrations. All LDL-PG binding values were divided by the plate control and expressed as an LDL-PG binding ratio. Fig 7 shows the LDL-PG binding ratio ranged from 0.6–4.6 in the control group and 1.7–5.8 in cardiovascular disease group. LDL-PG binding was significantly higher in adults with cardiovascular disease vs the control group ($p<0.05$) despite equal APOB concentrations.

## Discussion

We have developed and validated a robust, high throughput assay to measure LDL binding to the ECM of MOVAS cells. The binding is specific to LDL, mediated by sulfated GAGs and is sensitive to pH variations. This assay is reproducible with an intra- and inter-assay coefficient of variation of 2.9–7.2% and 9.4% respectively. In addition, the assay is able to detect differences in LDL-PG binding affinities among different LDL-sized particles and between adults with and without cardiovascular disease.

A variety of *in vitro* assays have been developed to detect LDL-PG binding as a marker of atherosclerotic risk [11, 13–17, 24]. These methods include insoluble complex formation, affinity chromatography, gel mobility shift assays, and ELISA assays. Most of these techniques involve the use of purified or extracted proteoglycan which are immobilized on solid matrices such as beads or plates before adding LDL. The bound LDL is then quantified using optical density, scintillation counting or antibodies against APOB. However, some of the methods can be labor-intensive, require specialized equipment, and pose technical challenges such as the use of radioactivity. To address these limitations, Lundstam *et al* [40] used decellularized ECM from confluent human arterial smooth muscle cells (HASMCs) to study LDL-PG binding. However, the decellularization process may alter the composition of individual proteoglycans and does not fully replicate the *in vivo* environment where cells are present. In contrast, our current assay using fixed vascular smooth muscle cells offers several advantages. First, there is no need for the lengthy and cumbersome process of GAG extraction, eliminating concerns of structural and functional alterations in the final product. By using a cell line that expresses several PGs, we can detect LDL binding to a combination of extracellular and cell surface PGs that have their structures preserved. Second, this cell-based assay includes non-PG fibrous proteins (such as collagen and fibronectin) that are present *in vivo*, providing a more representative experimental model.

Our immunofluorescent images (Fig 1) show that LDL favors binding to the ECM at a higher avidity than to cell surface where the LDL receptor and cell surface PGs are located in MOVAS cells. This observation is supported by several lines of evidence. Firstly, the expression of LDL receptors decreases with increasing cell density, suggesting a reduced number of LDL receptors in confluent MOVAS cells [46]. Secondly, human LDL is a very poor ligand for the mouse LDL receptor, having about an eighth of the binding ability compared to the human LDL receptor [47, 48]. Interestingly, greater LDL binding was observed in human ascending aorta vascular smooth muscle (HAAVAS) cells compared to MOVAS cells (S2 Fig in S1 File). This may be attributed to LDL binding to the human LDL receptor, but there is also a possibility that there are intrinsic species differences in proteoglycan composition [49], and differences in their affinities towards human LDL may also contribute. Thirdly, the LDL receptor-ligand interaction requires a buffer containing a minimum of 2 mM calcium [50], while our experiments used a calcium-free buffer (PBS, pH 7.4). Fourthly, LDL binding to the LDL receptor saturates at an LDL concentration of about 25 μg/mL. Higher LDL concentrations favor binding to the cell surface through sulfated PGs rather than the LDL receptor [51]. All our experiments were performed using mouse post-confluent MOVAS cells and human LDL dissolved in a calcium free buffer at concentrations well above 25 μg/mL. Therefore, we conclude that

most of the measured LDL was binding to the PGs either on the cell surface or the ECM and not to the LDL receptor.

In addition, *Galeano et al* [51] used LDL receptor negative skin fibroblasts from a patient with familial hypercholesterolemia and showed that LDL binds mainly to heparin sulfate proteoglycans (HSPGs) on the cell surface of these cells. When the cells were pretreated with heparinase (5 units/mL) and sodium chlorate (50 mM), LDL binding to the cells was reduced by 70% and 45% respectively, confirming that sulfated HSPGs were responsible for LDL binding to these cells. Our experiments with MOVAS cells also revealed a 56.2% reduction in LDL binding when cells were pretreated with 50 mM sodium chlorate (Fig 3D) and a 20% reduction with 1 unit/mL heparinase (data not shown). Similar results were obtained by *Olsson et al.* [52], who used normal human fibroblast pretreated with a mixture of heparinase, chondroitinase, and sodium chlorate. They also concluded that cell surface PGs contributed to LDL binding to the cell surface. Thus, we conclude that the majority of the LDL bound to the cell surface of MOVAS cells is binding to some form of PGs (HSPGs and CSPGs), indicating that the presence of cells does not significantly influence the results of this assay. Specifically, the cell-based assay system is comparable to an assay performed on plates coated with ECM GAGs as shown in Figs 3B, 4B and 5B.

We also investigated the impact of pH variations and LDL particle size on PG binding, which aligns with previous findings [51, 53]. The results revealed that LDL subfractions with smaller LDL particles exhibited a stronger affinity for PGs compared to fractions containing larger particles (Fig 6), and a similar trend was observed at lower pH (Fig 5). Specifically, at low pH (pH 4), LDL binding was higher compared to high pH (pH 7). Interestingly, Wang et al. [54] recently showed that acidification influences LDL particles by altering their size and cholesterol content. Under normal physiological conditions (pH 7.4), LDL particles are larger and contain more cholesterol while at pH 4, LDL particle size becomes smaller. It would be interesting to determine if a correlation exists between particle size and LDL-PG affinity in smaller LDL particles or if other factors play a role in their enhanced affinity.

Finally, our work also shows that LDL-PG binding is higher in adults with cardiovascular disease compared to those without (Fig 7). We hypothesize that this could be attributed to a greater number of individuals with small LDL particles in the cardiovascular disease group since these experiments were done at equal APOB concentration. However, we recognize that changes in LDL particle composition in patients with cardiovascular disease may also influence PG binding, and this aspect will be explored in future work. In future studies, we plan to evaluate this assay in other at-risk populations, to evaluate if it predicts future cardiovascular disease, and investigate whether LDL-PG binding can be reduced following interventions that improve cardiovascular risk, such as weight loss surgery. This will help determine the potential of this assay to assess higher or improved atherogenic risk among individuals over time.

One limitation of this study is our inability to identify the specific combination of PGs and non-PGs molecules responsible for LDL-ECM binding. Given the complex nature of the ECM, it is likely that non-PG components within the matrix or cells are involved. This conclusion is supported by our inhibition experiments, which suggest that approximately 56% of LDL binding could be inhibited by under-sulfation of PGs (Fig 4D), leaving about 44% of the bound LDL unaccounted for. Furthermore, our use of fixed cells as source of the matrix does not completely reflect the conditions observed in the arterial intima, where smooth muscle cells are embedded within the ECM and can migrate freely. Finally, we used a mouse cell line for this assay. While this could be viewed as a limitation, it is actually a strength of the current model. Human LDL binds poorly to the mouse LDL receptor, thus the measured LDL binding in the mouse cell line is attributed LDL binding to PG. This is unlike the human cell line where LDL binds both the LDL receptor and PGs. Given the concerns with the human LDL receptor,

high costs of maintaining human cell lines for repeated experiments over time, and data in human and mouse cell lines showing similar results, we show the mouse cell line is a valid model to study LDL-PG interactions.

In conclusion, we have developed an assay that allows for the quantification of LDL-PG binding using cells cultured in a 96- well format. This robust system enables us to efficiently screen patients or to study LDL-PG binding under various conditions and can easily be adapted for high throughput applications.

## Supporting information

**S1 File. LDL colocalizes with proteoglycans in human ascending aorta smooth muscle cells.**
(PDF)

**S2 File. Raw excel data.**
(ZIP)

**S1 Raw images. Original gel and blot images.**
(PDF)

**S2 Raw images. Fluorescent microscope images depicted in Fig 2.**
(ZIP)

## Author Contributions

**Conceptualization:** Debi K. Swertfeger, John T. Melchior, W. Sean Davidson, Amy Sanghavi Shah.

**Funding acquisition:** W. Sean Davidson, Amy Sanghavi Shah.

**Investigation:** Esmond N. Geh.

**Resources:** Hannah Sexmith, Anna Heink.

**Supervision:** Amy Sanghavi Shah.

**Validation:** Pheruza Tarapore.

**Writing – original draft:** Esmond N. Geh.

**Writing – review & editing:** Debi K. Swertfeger, W. Sean Davidson, Amy Sanghavi Shah.

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
