## [Decision Letter · Decision Letter 0]

3 Mar 2023

PONE-D-22-35204A Novel assay to measure low-density lipoproteins binding to proteoglycansPLOS ONE

Dear Dr. Geh,

Thank you for submitting your manuscript to PLOS ONE. After careful consideration, we feel that it has merit but does not fully meet PLOS ONE’s publication criteria as it currently stands. Therefore, we invite you to submit a revised version of the manuscript that addresses the points raised during the review process.

One of the reviewers pointed a major problem concerning the cell model. Your immunofluorescence images should be improved. Close-ups are necessary and also if you speak about colocalzation, confocal microscopy should be performed.

We look forward to receiving your revised manuscript.

Kind regards,

Boyan Grigorov

Academic Editor

PLOS ONE

Journal Requirements:

4. We note that you have stated that you will provide repository information for your data at acceptance. Should your manuscript be accepted for publication, we will hold it until you provide the relevant accession numbers or DOIs necessary to access your data. If you wish to make changes to your Data Availability statement, please describe these changes in your cover letter and we will update your Data Availability statement to reflect the information you provide

Reviewers' comments:

Reviewer's Responses to Questions

**Comments to the Author**

1. Is the manuscript technically sound, and do the data support the conclusions?

Reviewer #1: Partly

Reviewer #2: No

2. Has the statistical analysis been performed appropriately and rigorously? 

Reviewer #1: I Don't Know

Reviewer #2: No

3. Have the authors made all data underlying the findings in their manuscript fully available?

Reviewer #1: Yes

Reviewer #2: Yes

4. Is the manuscript presented in an intelligible fashion and written in standard English?

Reviewer #1: Yes

Reviewer #2: Yes

5. Review Comments to the Author

Reviewer #1: This study (Ms #: PONE-D-22-35204) sought to develop a novel, rapid, sensitive, and reproducible method for measuring low-density lipoproteins (LDLs) binding to proteoglycans (PGs) in combination of immunofluorescence microscopy with in-cell ELISA (ICE) technique (this technique is to determine LDL bound to PGs by quantifying ApoB). To test their method, mouse vascular smooth muscle (MOVAS) cells were used to produce PGs (extracted PGs and purified chondroitin sulfate or CS were also utilized), and LDLs were isolated from adults without (as a control) or with cardiovascular disease. By using the novel method, the authors confirmed the previously reported result that the LDLs from adults with cardiovascular disease have a higher affinity to PGs than those from healthy adults. LDL affinity to extracellular matrix (ECM; e.g., PGs) is a very important step in the initiation and development of atherosclerosis according to the retention hypothesis of atherogenesis. Therefore, any efforts to develop methods for detecting LDL-ECM/PGs affinity or interaction are encouraged and applausive. This manuscript can be considered for publication after addressing my following comments:

1. The background, applications (particularly its application in LDL and LDL-ECM/GP interaction if available), and advances of the ICE technique, should be described in the “Introduction” section. This description will help emphasize the novelty of this study.

2. Lines 83-85: The detailed information on the ultracentrifugation (e.g., speed, temperature, and time) should be provided. The abbreviation “UC-LDL” should be defined at the firm mention. Line 130: I suggest to remove the abbreviation UC and replaced with its full name (ultracentrifugation?).

3. In the “Materials and methods” section, the “Study participants” part should be moved to the location before or after the “lipoprotein isolation” part because the lipoproteins were isolated from the participants.

4. Line 125: There are two “overnight”.

5. Fig. 1: Based on the displayed 2-D fluorescence images, it is hard to determine whether the fluorescences (or molecules, e.g., ApoB and CS) are located pericellularly (or on the cell surface) or within the ECM or in cells. According to my experience, Fig. 1 (particularly the cells indicated by the white arrows) shows that the CS fluorescence is located mainly inside cells and that the ApoB fluorescence is located pericellularly (or on the cell surface). The 3-D fluorescence images should be provided to determine the location/co-localization of ApoB/CS. By the way, the legend to Fig. 1D,H is missing.

6. Besides CS, other types of PGs (e.g., heparin and dermatan sulfates) should also be detected by immunofluorescence imaging to determine CS is the major type of PGs produced by MOVAS cells and the major factor responsible for the LDL binding. If other types of PGs also contribute to the binding of LDL, the evidence from the experiments only testing LDL-CS interaction/co-localization is not solid enough to support the conclusions in this study.

7. Lines 208-209: Fig. 2A shows that LDL binds to MOVAS cells in a dose dependent manner which saturated at about 1 mg/mL. I cannot see the saturated concentration of LDL based on Fig. 1A (it seems that the bound LDL is still increasing at 2 mg/mL).

8. In this study, the authors used chondroitin sulfate antibody (alpha-CS) to block the CS-LDL interaction sites. However, it seems that the CS-LDL interaction sites (electrostatic interaction) are perhaps not overlapped with the CS-alpha-CS interaction sites (antigen-antibody interaction). Perhaps, a better interaction-blocking experiment is to block the negative charge of CS?

9. Fig. 4: The authors found that low pH enhances LDL binding to ECM/PGs/CS. However, a question may arise whether the influence of pH on the biomechanical properties of LDL particles also contributed to the changes in LDL binding to ECM/PGs/CS? This question should be addressed in the “Discussion” section. For example, it has been previously reported that low pH could not change the stickiness of LDL particles (Ref. 1) although the oxidation could increase the stickiness of LDL particles (Ref. 2). [Refences: (1) Wang K. etc. AFM detects the effects of acidic condition on the size and biomechanical properties of native/oxidized low-density lipoprotein. Colloids and Surfaces B: Biointerfaces. 2021. 208: 112053; (2) Wang K. et al. Dynamic AFM detection of the oxidation-induced changes in size, stiffness, and stickiness of low-density lipoprotein. Journal of Nanobiotechnology. 2020. 18: 167].

10. Line 265: The abbreviation CV should be defined at the first mention. All abbreviations in Table 1 should be defined.

11. Legend to Fig. 5: (a) The abbreviations CBB and FPLC should be defined; (b) Fig. 5D-F is missing; (c) The numbers 1-5 (or I-IV) on each graph should also be defined.

12. It will be better to compare the current method with other existing methods in the “Discussion” section based on the experimental data.

13. The authors should carefully check the whole “Discussion” section to correct potential typos and gramma errors.

Reviewer #2: This study by Geh and coworkers aims at designing a robust and reproducible assay to measure the binding of low-density lipoproteins (LDL) to proteoglycans, in view of a screening of blood samples from a population at risk of atherosclerosis.

Although the idea is elegant, this study has a major flaw. Indeed, the core of proteoglycans harbour

highly specific sugar molecules (GAG), chemically-modified at several places of the sugar ring (epimerisation, phosphorylation, sulfatation...). This high level of specificity is species-dependent (as pinpointed in Nikpour et al Glycobiology 2021, only to cite one). In consequence, using a mouse cell line to address such a question does not properly recapitulate the proteoglycan diversity in terms of GAG of a human organ/tissue.

Experiments should then be performed on a human cell line.

Minor point.

With a p value < 0.05, the method should not be considered as "highly sensitive".

6. PLOS authors have the option to publish the peer review history of their article (what does this mean?). If published, this will include your full peer review and any attached files.

Reviewer #1: No

Reviewer #2: No

---

## [Author Response · Author response to Decision Letter 0]

25 Jul 2023

Dear Editor and Reviewers, 

We greatly appreciate the editor and reviewers’ comments and suggestions in reviewing our manuscript entitled. We have carefully considered the comments and recommendations and made corresponding revisions to the manuscript. Below are our responses to each specific points.

Editor Comment.

One of the reviewers pointed a major problem concerning the cell model. Your immunofluorescence images should be improved. Close-ups are necessary and also if you speak about colocalization, confocal microscopy should be performed. 

As suggested, we conducted an experiment using a human cell line (human ascending aorta smooth muscle) and we have included confocal microscopy images that show colocalization of the LDL with chondroitin and heparan sulfate in both mouse and human cell lines. 

Reviewer 1.

1. The background, applications (particularly its application in LDL and LDL-ECM/GP interaction if available), and advances of the ICE technique, should be described in the “Introduction” section. This description will help emphasize the novelty of this study”

As recommended, this important bit of information has been added to the “Introduction” section of revised manuscript (lines 60-64). 

2. Lines 83-85: The detailed information on the ultracentrifugation (e.g., speed, temperature, and time) should be provided. The abbreviation “UC-LDL” should be defined at the firm mention. Line 130: I suggest to remove the abbreviation UC and replaced with its full name (ultracentrifugation?).

The revised manuscript now includes the information on speed, temperature, and time in the "Materials and Methods" section (lines 100). Additionally, we defined the abbreviation "UC-LDL" at its first mention and replaced UC in line 150. 

3. In the “Materials and methods” section, the “Study participants” part should be moved to the location before or after the “lipoprotein isolation” part because the lipoproteins were isolated from the participants.

As suggested, the "Study participants" part, has been moved just before “lipoprotein isolation” (see line 88),

4. Line 125: There are two “overnight”.

We have corrected this duplication (see line 144).

5. Fig. 1: Based on the displayed 2-D fluorescence images, it is hard to determine whether the fluorescences (or molecules, e.g., ApoB and CS) are located pericellularly (or on the cell surface) or within the ECM or in cells. According to my experience, Fig. 1 (particularly the cells indicated by the white arrows) shows that the CS fluorescence is located mainly inside cells and that the ApoB fluorescence is located pericellularly (or on the cell surface). The 3-D fluorescence images should be provided to determine the location/co-localization of ApoB/CS. By the way, the legend to Fig. 1D,H is missing.

To address this concern, we now include confocal microscopy images showing colocalization of APOB together with CS and HS in MOVAS (Fig 1), and human aortic vascular smooth muscle cells (supplemental S1 Fig) in the manuscript. The missing figure legends have also been included. 

6. Besides CS, other types of PGs (e.g., heparin and dermatan sulfates) should also be detected by immunofluorescence imaging to determine CS is the major type of PGs produced by MOVAS cells and the major factor responsible for the LDL binding. If other types of PGs also contribute to the binding of LDL, the evidence from the experiments only testing LDL-CS interaction/co-localization is not solid enough to support the conclusions in this study.

We now include confocal images of both heparin sulfate and chrondrotin sulfate. As CS has the highest affinity, in terms of binding to LDL than other proteoglycans, we have focused on CS but acknowledge other PGs may contribute to binding. (Reference: Mourão PA, Pillai S, Di Ferrante N. The binding of chondroitin 6-sulfate to plasma low density lipoprotein. Biochim Biophys Acta. 1981 May 5;674(2):178-87.)

7. Lines 208-209: Fig. 2A shows that LDL binds to MOVAS cells in a dose dependent manner which saturated at about 1 mg/mL. I cannot see the saturated concentration of LDL based on Fig. 1A (it seems that the bound LDL is still increasing at 2 mg/mL).

Agree. We corrected this sentence by deleting the “which saturated at about 1 mg/mL” (lines 235). 

8. In this study, the authors used chondroitin sulfate antibody (alpha-CS) to block the CS-LDL interaction sites. However, it seems that the CS-LDL interaction sites (electrostatic interaction) are perhaps not overlapped with the CS-alpha-CS interaction sites (antigen-antibody interaction). Perhaps, a better interaction-blocking experiment is to block the negative charge of CS?

We agree. The CS-LDL interaction is an electrostatic interaction between positively charged LDL and negatively charged sulfate groups of CS. For this reason, in addition to antibody inhibition, we performed an endogenous inhibition of sulfation using sodium chlorate which is a 3′-phosphoadenosine 5′-phosphosulfate (PAPS) synthetase inhibitor (Fig 6D). Thus, reducing or indirectly blocking the amount of available sulfate groups not only on CS but also on other sulfate containing PGs.

9. Fig. 4: The authors found that low pH enhances LDL binding to ECM/PGs/CS. However, a question may arise whether the influence of pH on the biomechanical properties of LDL particles also contributed to the changes in LDL binding to ECM/PGs/CS? This question should be addressed in the “Discussion” section. For example, it has been previously reported that low pH could not change the stickiness of LDL particles (Ref. 1) although the oxidation could increase the stickiness of LDL particles (Ref. 2). [Refences: (1) Wang K. etc. AFM detects the effects of acidic condition on the size and biomechanical properties of native/oxidized low-density lipoprotein. Colloids and Surfaces B: Biointerfaces. 2021. 208: 112053; (2) Wang K. et al. Dynamic AFM detection of the oxidation-induced changes in size, stiffness, and stickiness of low-density lipoprotein. Journal of Nanobiotechnology. 2020. 18: 167].

We have made changes in the revised manuscript “Discussion” section to address this concern (see lines 398-402). In addition, we have added Ref 1 to the discussion section.

10. Line 265: The abbreviation CV should be defined at the first mention. All abbreviations in Table 1 should be defined.

The abbreviations have been defined in line 296. 

11. Legend to Fig. 5: (a) The abbreviations CBB and FPLC should be defined; (b) Fig. 5D-F is missing; (c) The numbers 1-5 (or I-IV) on each graph should also be defined.

All the above changes have been made in the revised manuscript (see lines 320-321)

12. It will be better to compare the current method with other existing methods in the “Discussion” section based on the experimental data.

We think that comparing methods based on experimental data could be challenging because of the variations in experimental protocols, specific antibodies and detection methods used. Additionally, most ELISA based techniques generally report lower limits of detection that vary 0.1 �g/mL to 20 �g/mL using a buffer that has a pH of about 5.5. In our case, we chose to use a physiologically relevant buffer, PBS at pH 7.4, and our limit of detection was about 0.5 �g/mL (Supplemental S4 Fig) so direct comparisons are challenging. We acknowledge existing methods with corresponding references. 

13. The authors should carefully check the whole “Discussion” section to correct potential typos and gramma errors.

The "Discussion" section has been revised to address grammatical errors and typos. Furthermore, new information has been incorporated into this section, which was not available at the time of submission. Some sentences that did not align with the discussion have been removed.

Reviewer 2.

1. Although the idea is elegant, this study has a major flaw. Indeed, the core of proteoglycans harbour highly specific sugar molecules (GAG), chemically-modified at several places of the sugar ring (epimerisation, phosphorylation, sulfatation...). This high level of specificity is species-dependent (as pinpointed in Nikpour et al Glycobiology 2021, only to cite one). In consequence, using a mouse cell line to address such a question does not properly recapitulate the proteoglycan diversity in terms of GAG of a human organ/tissue. Experiments should then be performed on a human cell line.

We appreciate your suggestion (see line 374). However, utilizing a human cell line expressing the LDL receptor presents challenges due to the binding of both human LDL and PGs and to the LDL receptor. We were unable to locate a human cell line that lacks the LDL receptor, particularly a vascular smooth muscle cell line. However, experiments were repeated in a human ascending aorta smooth muscle cells (HAAVAS) in parallel with the MOVAS cell line (supplemental S2 Fig). LDL binding in HAAVAS was over three times higher than in MOVAS, and we attributed a portion of this binding to the LDL receptor. Since the human LDL binds poorly to the mouse LDL receptor, most of the binding in the mouse cell line is attributed to PG binding. Maintenance cost of human cell lines is significantly higher, approximately 40 times more, making it financially impractical for widespread use and repeated experiments over time. Given the concerns with the human LDL receptor, high experimental costs, and data in human and mouse cell lines showing similar results, we believe that using the mouse cell line is a valid model to study LDL-PG interactions. 

2. With a p value < 0.05, the method should not be considered as "highly sensitive".

We have softened the language in the revised manuscript (see line 30).

---

## [Decision Letter · Decision Letter 1]

4 Sep 2023

A novel assay to measure low-density lipoproteins binding to proteoglycans

PONE-D-22-35204R1

Dear Dr. Geh,

We’re pleased to inform you that your manuscript has been judged scientifically suitable for publication and will be formally accepted for publication once it meets all outstanding technical requirements.

Kind regards,

Boyan Grigorov

Academic Editor

PLOS ONE

Additional Editor Comments (optional):

Reviewers' comments:

Reviewer's Responses to Questions

**Comments to the Author**

1. If the authors have adequately addressed your comments raised in a previous round of review and you feel that this manuscript is now acceptable for publication, you may indicate that here to bypass the “Comments to the Author” section, enter your conflict of interest statement in the “Confidential to Editor” section, and submit your "Accept" recommendation.

Reviewer #1: All comments have been addressed

Reviewer #2: All comments have been addressed

2. Is the manuscript technically sound, and do the data support the conclusions?

Reviewer #1: Yes

Reviewer #2: Yes

3. Has the statistical analysis been performed appropriately and rigorously? 

Reviewer #1: I Don't Know

Reviewer #2: Yes

4. Have the authors made all data underlying the findings in their manuscript fully available?

Reviewer #1: Yes

Reviewer #2: Yes

5. Is the manuscript presented in an intelligible fashion and written in standard English?

Reviewer #1: Yes

Reviewer #2: Yes

6. Review Comments to the Author

Reviewer #1: Most of my concerns have been addressed. Thank the authors for their contribution in this research field.

Reviewer #2: (No Response)

7. PLOS authors have the option to publish the peer review history of their article (what does this mean?). If published, this will include your full peer review and any attached files.

Reviewer #1: No

Reviewer #2: No

---

## [Editor Report · Acceptance letter]

15 Sep 2023

PONE-D-22-35204R1 

A novel assay to measure low-density lipoproteins binding to proteoglycans 

Dear Dr. Geh:

I'm pleased to inform you that your manuscript has been deemed suitable for publication in PLOS ONE. Congratulations! Your manuscript is now with our production department. 

Kind regards, 

on behalf of

Dr. Boyan Grigorov 

Academic Editor

PLOS ONE